# Masked Attention as a Mechanism for Improving Interpretability of Vision Transformers

**Clément Grisi**                                                                     CLEMENT.GRISI@RADBOUDUMC.NL
**Jeroen van der Laak**
**Geert Litjens**
*Computational Pathology Group, Radboudumc, Netherlands*

## Abstract

Vision Transformers are at the heart of the current surge of interest in foundation models for histopathology. They process images by breaking them into smaller patches following a regular grid, regardless of their content. Yet, not all parts of an image are equally relevant for its understanding. This is particularly true in computational pathology where background is completely non-informative and may introduce artefacts that could mislead predictions. To address this issue, we propose a novel method that explicitly masks background in Vision Transformers' attention mechanism. This ensures tokens corresponding to background patches do not contribute to the final image representation, thereby improving model robustness and interpretability. We validate our approach using prostate cancer grading from whole-slide images as a case study. Our results demonstrate that it achieves comparable performance with plain self-attention while providing more accurate and clinically meaningful attention heatmaps.

**Keywords:** Vision Transformers, attention, computational pathology, prostate cancer

## 1. Introduction

Adoption of whole-slide imaging technology in pathology has contributed to the growing availability of large digitized datasets. This shift towards digital pathology has fostered computer vision research to support and augment pathologists with deep learning algorithms. However, conventional deep learning methods are ill-equipped to handle the enormous sizes of whole-slide images (WSIs), which usually exceed the memory capacity of graphics processing units. A popular approach to overcome this memory bottleneck involves partitioning WSIs into smaller, more manageable patches (Campanella et al., 2019). This technique aligns well with the operational mechanics of Vision Transformers (ViTs) (Dosovitskiy et al., 2021), sparking increased interest in their application within computational pathology (Shao et al., 2021; Chen et al., 2022, 2024).

While ViTs process images by breaking them into smaller patches following a regular grid, they do not account for the fact that all parts of an image are not equally relevant or informative. Uniform background areas are often less informative than more cluttered, dense areas. In computational pathology, background is not only low-informative but fundamentally devoid of any diagnostic value. Including tokens that correspond to background areas may introduce artefacts that could mislead predictions and compromise model interpretability, possibly resulting in clinically irrelevant hotspots in attention maps. To address this issue, we propose a simple method that explicitly masks background in the attention

mechanism of ViTs. By doing so, we ensure tokens corresponding to background do not contribute to the final image representation. Omitting visually present but diagnostically irrelevant information should not only sharpen the signal-to-noise ratio, but also result in attention heatmaps that are both more visually coherent and easier to interpret.

## 2. Proposed Method

**Hierarchical Vision Transformer.** The inherent hierarchical structure within whole-slide images spans across various scales, from tiny cell-centric regions containing fine-grained information, up to the entire slide which exhibits the overall intra-tumoral heterogeneity of the tissue microenvironment. Drawing inspiration from this layered structure, our model consists of a Hierarchical Vision Transformer that processes whole-slide images at three nested scales (Grisi et al., 2023). Slides are unrolled into non-overlapping $2048 \times 2048$ regions, capturing macro-scale interactions between clusters of cells. These are further unrolled into non-overlapping $256 \times 256$ patches, depicting cell-to-cell interactions. A pre-trained ViT-S/16 is used to embed these patches into feature vectors. Then, a second Transformer aggregates the representations of $256 \times 256$ patches within larger $2048 \times 2048$ regions. Finally, a third Transformer pools region-level tokens into a slide-level representation that is projected to class logits for loss computation (Appendix A, Figure 2).

**Masked Attention.** When extracting regions from whole-slide images, only those containing tissue are retained as fully background regions contain no informative content. However, when these regions are further unrolled into non-overlapping $256 \times 256$ patches, some patches may still contain no tissue (Appendix C, Figure 4). To ensure that region-level representations are exclusively derived from patches containing tissue, we propose a novel *masked attention* method. By leveraging fine-grained tissue segmentation masks, our approach explicitly nullifies the contribution of entirely background patches during self-attention, thereby enhancing the quality of extracted features. We provide a pseudo code implementation in Appendix B.

## 3. Experimental Results

**Dataset.** To assess the robustness of the proposed method, we use the PANDA dataset (Bulten et al., 2022). It is the largest publicly available dataset of H&E stained prostate WSIs to date, with $11,554$ prostate biopsies curated from two different sites. All slides are provided at a pixel spacing close to $0.50~\mu$m, together with their ISUP score (Appendix D).

**Data Preprocessing & Evaluation Metric.** We used an internally developed model to automatically segment tissue in each slide, from which we extract non-overlapping $2048 \times 2048$ regions at the resolution closest to $0.50~\mu$m (Appendix C). We split PANDA development set into 5 cross-validation folds, stratifying on the ISUP score. To evaluate the model's classification performance, we report averaged quadratic weighted kappa scores on the tuning set, as well as on the combined public and private test sets.

**Prostate Cancer Grading.** For each fold, we pretrain the first Transformer on the training set via the student-teacher knowledge distillation framework DINO (Caron et al., 2021). This Transformer is used as a feature extractor to embed each slide into a $(M_i^{2048}, 64, 384)$

feature vector, where $M_i$ stands for the number of $2048 \times 2048$ regions extracted in the $i$-th slide. The last two Transformers are then jointly trained to map these sequences to ISUP scores. We formulate the classification problem as a regression task and use the Mean Squared Error loss to leverage the ordinal nature of the ISUP scores. Classification results are summarized in Table 1. Masked self-attention achieves comparable performance with plain self-attention.

| Attention Mechanism | Tune Score | Combined Test Score |
|---|---|---|
| Plain self-attention | $0.945 \pm 0.003$ | $0.899 \pm 0.008$ |
| Masked self-attention | $0.946 \pm 0.003$ | $0.899 \pm 0.009$ |

Table 1: ISUP score classification results. We report quadratic weighted kappa, averaged over the 5 cross-validation folds.

**Model Interpretability.** Attention heatmaps offer a streamlined form of model interpretability by revealing the specific image features that the model has learned to associate with particular classes. Figure 1 shows attention heatmaps for the region-level Transformer. While some background patches display high attention values in plain self-attention heatmaps (Figure 1($b$)), all background patches are given no attention in masked self-attention heatmaps (Figure 1($c$)). Additional visualizations at the slide level are provided in Appendix E.

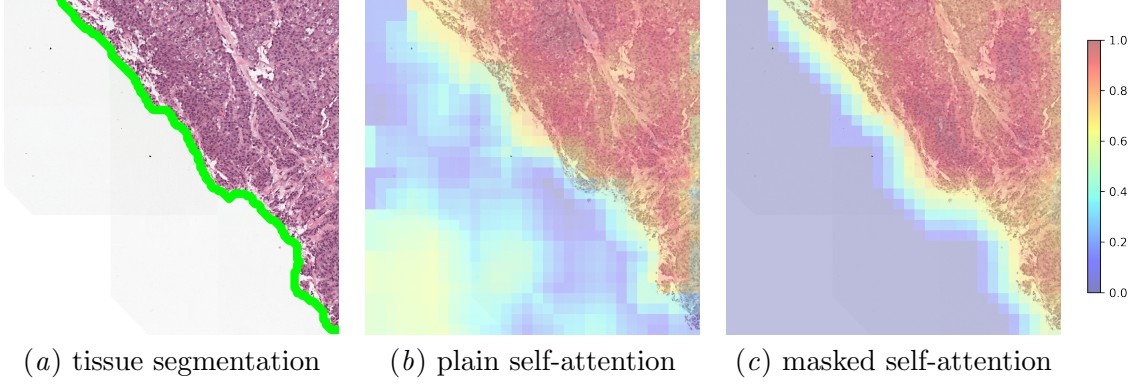

(a) tissue segmentation     (b) plain self-attention     (c) masked self-attention

Figure 1: Region-level attention maps.

## 4. Conclusion

In conclusion, our proposed masked attention strategy improves model interpretability by explicitly excluding irrelevant patches from contributing to self-attention in Vision Transformers. This approach is particularly beneficial in computational pathology where the inclusion of non-informative background content can introduce artefacts that can compromise model reliability. Our results demonstrate that masked attention achieves comparable performance with plain self-attention while providing more accurate and clinically meaningful heatmaps. This method has the potential to enhance the accuracy, robustness, and

interpretability of ViT-based models in digital pathology, ultimately contributing to improved diagnostic accuracy.

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

## Appendix A. Architecture Overview

Figure 2 shows the multi-stage Hierarchical Vision Transformer architecture we use in this work. It features three Vision Transformers, followed by a simple linear classifier that projects the slide-level embedding onto the desired number of classes.

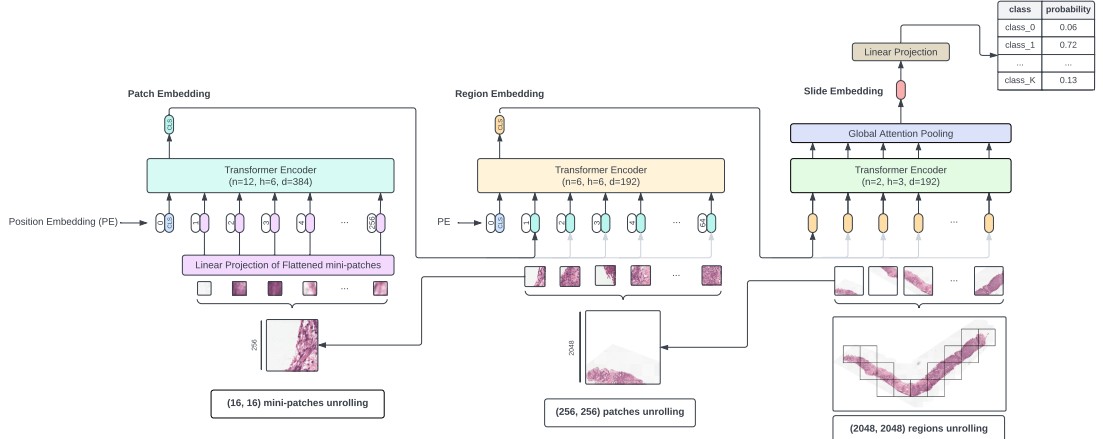

Figure 2: Overview of our Hierarchical Vision Transformer for whole-slide image analysis. This figure illustrates the multi-scale processing of whole-slide images.

## Appendix B. Masked Attention Pseudo Code

The *masked attention* module expects the input sequence $x$ – of shape $(M_i^{2048}, 64, 384)$ – as well as a *pct* tensor of shape $(M_i^{2048}, 1, 64)$ containing the tissue percentage for each $256 \times 256$ patch within each $2048 \times 2048$ regions in a slide.

---

**Algorithm 1:** Nullifying the contribution of background patches

---

**Input:** $x$ of shape (M, 64, 384), *pct* of shape (M, 1, 64)
**Output:** $x_{\text{attended}}$, the attended tensor
$q$, $k$, $v \leftarrow$ self.qkv($x$)
raw_attn $\leftarrow$ ($q$ @ $k$.T ) * scale
$pct \leftarrow pct$.unsqueeze(1).expand(-1, self.num_heads, -1, -1)
masked_attn $\leftarrow$ raw_attn.masked_fill($pct$ == 0, float("-inf"))
attn $\leftarrow$ masked_attn.softmax(dim=-1)
$x_{\text{attended}} \leftarrow$ (attn @ $v$).T

---

The full code is publicly available at github.com/computationalpathologygroup/hvit.

## Appendix C. Data Preprocessing

Figure 3 shows an example result of our tissue segmentation and region extraction algorithm. Due to potential tissue segmentation irregularities, regions containing fewer than 10% tissue were discarded.

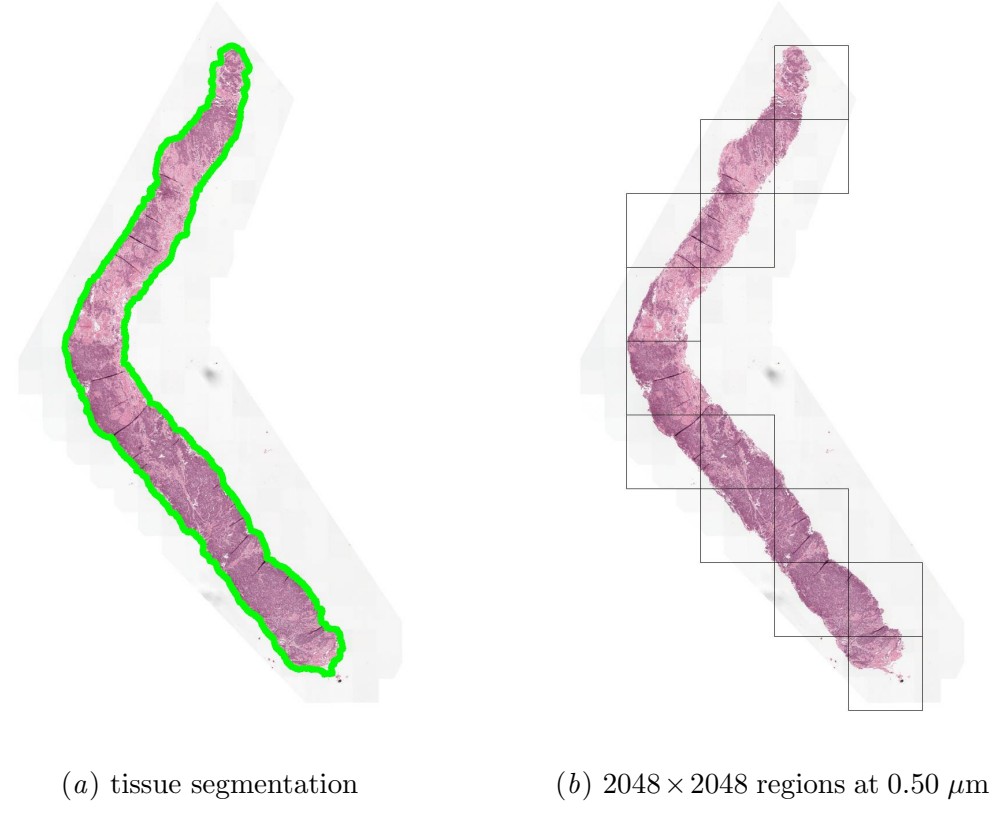

$(a)$ tissue segmentation $\qquad$ $(b)$ $2048 \times 2048$ regions at $0.50$ $\mu$m

Figure 3: Example result of data preprocessing pipeline

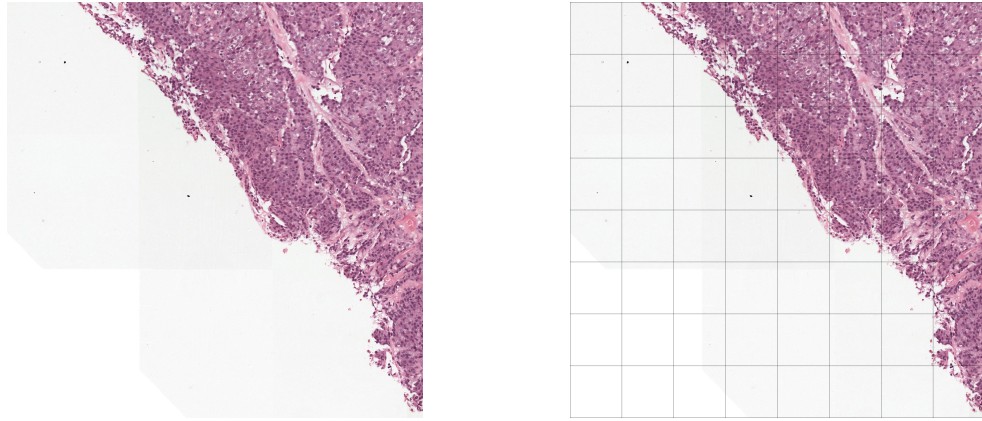

Figure 4: Unrolling a $2048 \times 2048$ region into non-overlapping $256 \times 256$ patches

## Appendix D. PANDA Dataset Details

In Table 2, we provide a summary of the main characteristics of the PANDA dataset.

Table 2: PANDA dataset summary

| Center | Scanner | Spacing ($\mu$m) | # dev | # public test | # private test |
|---|---|---|---|---|---|
| Radboud | 3DHistech | 0.48 | 5160 | 195 | 333 |
| Karolinska | Leica | 0.50 | 2193 | 97 | 150 |
| Karolinska | Hamamatsu | 0.45 | 3263 | 101 | 62 |

Pathologists classify tumors into different growth patterns by analyzing the histological architecture of the tumor tissue. Tissue specimens are then categorized into one of five groups based on the distribution of these patterns in the tumor. Figure 5 shows the grade group distribution for the development set, the public test set and the private test set.

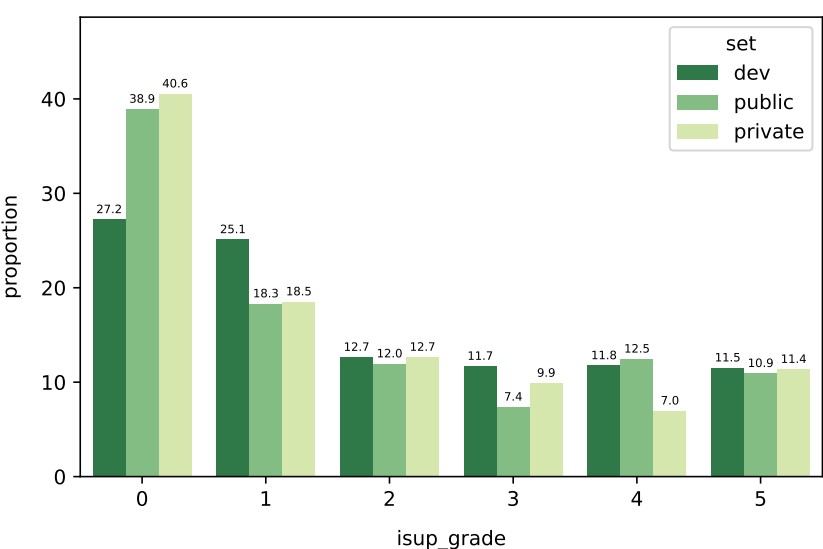

Figure 5: PANDA label distribution

## Appendix E. Stitched Attention Heatmaps

Stitched attention heatmaps provide a comprehensive visualisation of the model attention, offering a more intuitive understanding of which parts of the slide contribute most significantly to the model's decision-making process.

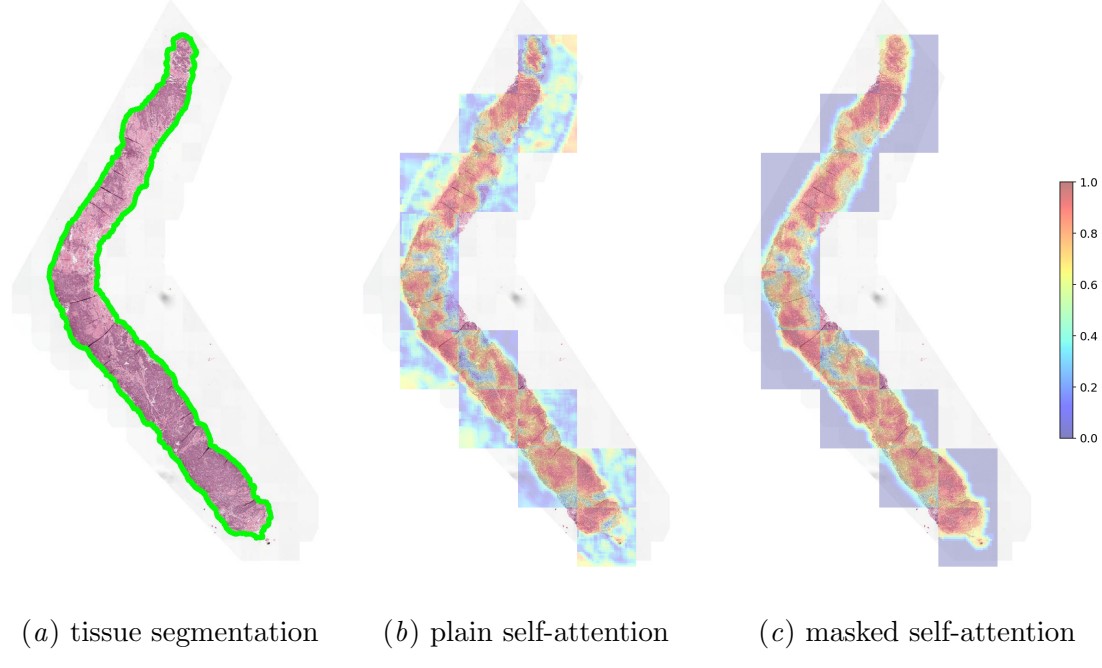

$(a)$ tissue segmentation $\qquad$ $(b)$ plain self-attention $\qquad$ $(c)$ masked self-attention

Figure 6: Stitched region-level attention heatmaps

