# OpenReview forum: "Masked attention as a mechanism for improving interpretability of Vision Transformers"
_MIDL.io/2024/Short_Papers — MIDL 2024 Short Papers_

### Official Review · Reviewer_NWot · 2024-04-22

**Confidence:** 2
**Final Rating:** 3.5

**Review:**

Summary
-------

The paper proposes a hierarchical transformer network for prostate cancer grading on whole-slide histopathology images. The hierarchical structure allows multi-resolution processing of pathology images. The authors propose a "masked attention" approach to prevent empty background regions from contributing to the attention maps.

This work was apparently already accepted to the Medical Imaging meets NeurIPS 2023 workshop (non-archival).


Strengths
---------

- The proposed hierarchical approach seems useful for histopathology data, and it would be interesting to see a more thorough evaluation in a longer version of the paper, should the authors decide to extend the work.


Weaknesses
----------

- The body of the paper is not really self-contained, and the authors makes quite liberal (borderline) use of the appendix. I would leave it to the program chairs to decide whether that's a problem.

- A more in depth analysis of the attention heat maps would have been interesting, as this seems to be one of the central aims of the paper.

- I do not understand the difference between Fig. 1c and 1d. Was the heatmap simply masked in a post processing step? What is the purpose of this?

---

### Decision · Program_Chairs · 2024-04-26

Accept